# Analysis of medico-legal claims related to deliveries: Caesarean section vs. vaginal delivery

Nasim Eshraghi[1], Marjan Ghaemi[1], Zahra Shabannejad[2], Elham Bazmi[2], Mehdi Foroozesh[2], Mohammad Haddadi[1], Sepideh Azizi[1], Zeinab Mansouri[1]*, Sedigheh Hantoushzadeh[1]*

1 Vali-E-Asr Reproductive Health Research Center, Family Health Research Institute, Tehran University of Medical Sciences, Tehran, Iran, 2 Legal Medicine Research Center, Iranian Legal Medicine Organization, Tehran, Iran

* hantoushzadeh@tums.ac.ir (SH); Znb.mansouri@gmail.com (ZM)

**Data Availability Statement:** All relevant data are within the paper and its Supporting information files.

## Abstract

### Background

The Iranian National Health Service (NHS) suggested that gynecologists face a higher risk of medicolegal claims, with a significant number of claims being related to delivery events. This study aimed to investigate the factors associated with delivery related claims.

### Method

In this cross-sectional study, we conducted an analysis of medico-legal documents which related to complications during delivery events and presented to Iranian Medical Legal Organization spanning from March 2018 to February 2020. A total of 227 legal prosecutions that were initiated by patients or, in cases where that wasn't possible, by their families, were included in the study and all of them were evaluated in commission with experienced professionals. The data collection phase occurred between February 2023 and May 2023. The collected data encompassed various aspects, including patient characteristics mode of delivery, reasons for claims, hospital type, accused party, the occurrence of instrumental delivery and the final disposition of the claims (paid claims or closed claims). Paid claims represent successful lawsuits where the healthcare provider or their insurer made a financial settlement to the patient. Closed claims encompass those that were either denied or dismissed. Chi-square or t-tests were employed to compare factors between paid claims and closed claims.

### Result

In this study, it was observed that vaginal delivery was performed in 51.1% of the claims, whereas 48.9% underwent a caesarean section.. Approximately half of the claims were against obstetrician-gynecologists, and 33% of the claims against other providers were against midwives.. The majority of complaints were related to perinatal mortality (34.8%) and neonatal asphyxia (18.5%). In 58.1% of cases, no malpractice was identified, while

**Funding:** The author(s) received no specific funding for this work.

**Competing interests:** The authors have declared that no competing interests exist.

41.9% resulted in paid claims. Also, there were no significant differences between the paid claims and closed claims groups in several factors, such as the type of hospital (P = 0.904), maternal age (P = 0.157), type of delivery (P = 0.080), and accused party (P = 0.168). However, the number of instrumental deliveries (13.8% of vaginal deliveries) and the reasons for claims, exhibited significant differences between the two claims (P = 0.021, P<0.001 respectively).

## Conclusion

This study found that maternal complications were more common in caesarean sections, while neonatal claims were more prevalent in vaginal deliveries. The study recommended public health interventions to reduce the overall prevalence of delivery-related claims.

## Introduction

Obstetricians and gynecologists face higher risks of medicolegal claims and compensation payments compared to most other physicians. This has been reported in countries such as Iran, Egypt and Saudi Arabia [1–3]. According to a report by the United Kingdom National Health Service (NHS), the field of obstetrics has consistently experienced the highest number of litigations over the past decade [4]. This increased risk can be attributed to the high-risk nature of obstetric procedures, as well as the emotionally charged aspects of the specialty. Factors such as unpredictable outcomes, communication challenges, and the sensitive nature of reproductive health all contribute to the heightened likelihood of malpractice claims within this field [5–7]. Recent reports indicate that more 60% of medicolegal claims in the field of obstetrics are specifically related to obstetrical issues. Furthermore, it has been observed that around 70% of these obstetric claims pertain to events occurring during the delivery process [6, 8].

The impact of caesarean section delivery on the frequency of disputes against obstetricians is a subject of debate [9]. It has been reported that caesarean section delivery is associated with an increased risk of complications, including maternal post-operative infection, febrile responses leading to maternal morbidity, extended hospital stays, and psychosocial outcomes that affect the mother-infant relationship. Additionally, it may contribute to difficulties in initiating successful breastfeeding and raise the likelihood of adverse neonatal outcomes [10, 11]. As a result, these complications can potentially result in a rise in medicolegal claims [12].

Nevertheless, it is crucial to note that caesarean delivery has been shown to reduce maternal mortality rates in most countries, but the reduction occurred when the rate of caesarean was between 5–10% [13]. Some studies have indicated that there is no significant difference between caesarean section and vaginal delivery groups regarding markers of birth asphyxia, Apgar scores below seven at five minutes, neonatal seizures, and hypoxic ischemic encephalopathy [14].

The global average rate of caesarean section from 2010 to 2018 was approximately 21% [15]. In Iran, the caesarean section rate is approximately 52%, over two times higher than the global average [16]. Various factors contribute to this trend, including experiences of infertility, considerations of fetal size, previous traumatic childbirth experiences, recommendations from physicians, and fear of vaginal trauma [17]. The increased prevalence of caesarean section births signifies a significant turning point in the realms of medicine, medical ethics, and forensic medicine.

However, precise statistics regarding maternal and neonatal complications and the associated claims against physicians after vaginal births and caesarean sections are not readily available. Nevertheless, the objective of our study was to examine medicolegal documents pertaining to delivery events. Through the analysis these documents, our aim was to gain valuable insights into the reasons behind complaints associated with such delivery events.

## Methods

### Study design

This cross-sectional study was conducted using medico-legal dispute documents from March 2018 to February 2020 in Tehran, Iran. In the Iranian medico-legal framework, all patients or, if the patient is unable, their family members can file claims against healthcare providers. All relevant medical-legal documents pertaining to these malpractice claims are finally presented to the Iranian Medical Legal Organization. The commission at this organization, which includes 3 experienced professionals, conducts a detailed evaluation of each claim. The medical experts carefully assess all the documents, and all the healthcare providers involved are invited to present at the commission hearings. At the end of each session, the judges declare their decision. The patient or healthcare provider then have one month to protest the decision of the court. If no protest is filed, the decision is considered final. However, if either the patient or healthcare provider protests the decision, another commission with 5 professionals will be convened to re-evaluate the claim. This process can continue with commissions of 7, 9, and finally 11 professionals if the parties continue to dispute the ruling [3].

The final decision determines whether the claim should be closed (dismissed) or result in a financial payout to the plaintiff. Paid claims represent successful malpractice lawsuits where the healthcare provider or their insurer made a financial settlement to the patient. Closed claims encompass those that were denied or dismissed by the commission.

### Eligibility criteria

In this study, a total of 227 legal claims related to delivery events, including both caesarean sections and vaginal deliveries, along with their associated complications, were evaluated and reported. The data collection phase took place between February 2023 and May 2023. These cases were presented to the legal court in Tehran province over a period of three years. The inclusion criteria for the study were as follows: the referred complaints must be related to the delivery events provided in hospitals, clinics, and other healthcare facilities. Additionally, the referred cases must have been reviewed and concluded by the Medical Commission of this department. The exclusion criteria included cases that did not have a final decision by the Medical Commission, as well as those that did not have detailed information regarding the nature of the claims.

### Outcome measures

The data collection was conducted by reviewing medicolegal documents. The collected information encompassed various factors, including the patients' age, the mode of delivery (caesarean section—emergency, elective, or indicated—or vaginal delivery), instrumental delivery, the reasons for claims, the type of hospital (private, teaching, or non-academic public), and the accused party (such as obstetrician-gynecologists or other healthcare providers like neonatologists, anaesthetists, and midwives) and the final outcome of the claims (whether they were paid or closed). In private and governmental hospitals, the management of patients was carried out

by OB/GYN physicians. However, in teaching hospitals, OB/GYN residents under the supervision of OB/GYN assistant professors managed the patients.

The patients' claims were initially classified into two main categories: maternal and neonatal. Within the neonatal category, complications were further divided into subcategories including perinatal mortality, neonatal asphyxia, birth traumas, and neonatal respiratory distress syndrome (RDS). On the other hand, maternal complications were categorized as maternal death, maternal organ damage (such as bowel perforation, bladder damage, urethral damage, and uterine rupture), and maternal post-procedural complications (such as rectovaginal fistula and leakage, deep vein thrombosis (DVT), and hematoma).

## Statistical analysis

Statistical analysis was performed using SPSS 24.0 software. Categorical data were presented as numbers and percentages, while numerical data were displayed as means and standard deviations (SD) or median and interquartile range (IQR) depending on the distribution. The Chi-square or t-test was employed to compare variables between different groups. Additionally, a multivariable regression analysis was conducted to assess the relationship between each factor and the risk of paid claims. The significance level was set at a p-value of <0.05.

## Ethical approval

The study received ethical approval from Tehran University of Medical Sciences (TUMS) with the ethical number IR.TUMS.IKHC.REC.1401.145. Informed consent was obtained from all participants, and it is essential to emphasize that patient data were collected and presented in a manner that maintained confidentiality and privacy.

## Results

A total of 227 women with obstetric claims meeting the inclusion criteria were included in this study over a span of three years. The median age of the patients was 31 years, ranging from 18 to 43 years. In addition, 63 (287.8%) of the patients were primigravida. Among the cases, 60 (26.4%) were related to maternal problems, and 167 (73.6%) were related to neonatal complications.

The median time from delivery to the filing of a lawsuit was 6.7 months, with an interquartile range (IQR) of 3.1 to 15.0 months. Out of the total number of claims, 73 cases (32.3%) occurred in non-academic public hospitals, while 68 cases (30.0%) occurred in academic hospitals. Furthermore, 86 cases (37.9%) were attributed to private hospitals. The distribution of reasons for complaints against healthcare providers and the mode of delivery is shown in Table 1. The majority of complaints were related to perinatal mortality, accounting for 79 cases (34.8%), while neonatal accounted for 42 cases (18.5%). A total of 116 (51.1%) patients underwent normal vaginal delivery, and 111 (48.9%) cases underwent a caesarean section.

In 115 cases (50.6%), the prosecution was directed towards obstetricians and gynecology (OB/GYN) physicians, while in the remaining 112 cases (49.4%), it was reported against other medical staff members. The distribution of claims against other medical staff is presented in Table 2. Most of the complaints were related to midwives 37 (33%), followed by Pediatrics 22 (19.6%).

The data analysis revealed that in 132 (58.1%) cases, no malpractice occurred, and they were successfully defended. In the remaining 95 (41.9%) cases, claims were paid out. There were no significant differences observed between the two claims in terms of various factors, including maternal age, accused parties (OB/GYN physician vs. other medical staff), and type of hospital. While the occurrence of maternal complaints and the number of primigravid patients were higher in the claims associated with indemnity payment, these differences did

**Table 1. Information and types of deliveries of 227 claims.**

| Characteristics | |
|---|---|
| Maternal age (year) (median, IQR) | 31 (27–35) |
| **Gravidity** | |
| Primigravida | 63 (27.8) |
| Multigravida | 164 (72.2) |
| **The reasons for lawsuit** | |
| **Neonatal complications** | 167 (73.6) |
| Perinatal mortality | 79 (34.8) |
| Neonatal asphyxia | 42 (18.5) |
| Birth trauma | 41 (18.1) |
| Neonatal RDS | 3 (1.3) |
| other neonatal complications | 2 (0.9) |
| **Maternal problems** | 60 (26.4) |
| Maternal organ damage | 21 (9.3) |
| Maternal death | 19 (8.4) |
| Post procedure complications | 16 (7.0) |
| Other maternal problems | 4 (1.8) |
| The time between delivery to lawsuit (month) (media, IQR) | 6.7 (3.1–15) |
| **Mode of delivery** | |
| Vaginal delivery | 116 (51.1) |
| Emergency C/S | 71 (31.3) |
| C/S with indication | 29 (12.8) |
| Elective C/S | 11 (4.8) |
| **Instrumental delivery** | |
| Positive | 16 (13.8) |
| Negative | 100 (86.2) |
| **Type of hospital** | |
| Private hospital | 86 (37.9) |
| Governmental hospital | 73 (32.2) |
| Teaching hospital | 68 (29.9) |
| **Accuser** | |
| Gynecologist | 115 (50.7) |
| Other medical staffs | 112 (49.3) |

Data are presented as median and IQR or number and percentage (%)

IUFD: Intrauterine fetal demise, RDS: Respiratory distress syndrome, C/S: caesarean section, IQR: interquartile range

not reach statistical significance. It was observed that the proportion of paid claims was similar between women who underwent vaginal delivery and women who underwent an emergency caesarean section. This suggests that both claims faced a comparable risk of experiencing malpractice incidents (Table 3).

Despite the rate of elective caesarean section being lower (11, 4.8%) compared to other modes of delivery in legal claims, it is noteworthy that the highest rate (8 out of 11, 72.7%) of payment compensation outcomes was associated with elective caesarean sections. Instrumental delivery was found to be significantly higher in the paid claims compared to the closed claims (P = 0.021). Furthermore, when conducting a multivariable adjusted regression analysis, the study found that instrumental delivery was the only factor that increased the risk of paid claims,

**Table 2. The claims against other health care provider.**

| Accuse against other health care providers (n = 112) | Number (%) |
|---|---|
| Midwife | 37 (33) |
| Nurse | 11 (9.8) |
| Assistant (surgeon assistant or aesthetic assistant) | 20 (17.8) |
| Paediatric | 22 (19.6) |
| Anaesthesiologist | 7 (6.2) |
| General surgeon | 4 (3.6) |
| Internal medicine physician | 4 (3.6) |
| Emergency medicine Specialist | 2 (1.8) |
| Laboratory technical assistant | 4 (3.6) |
| Blood bank staff | 1 (0.9) |

Data are presented as number and percentage (%)

**Table 3. The comparison between paid claims and closed claims regarding malpractice occurrence or not during legal prosecution was conducted.**

| All claims (n = 227) | Paid claims (n = 95) n(%) | Closed claims (n = 132) n(%) | P value |
|---|---|---|---|
| Age | 31.39 ± 5.31 | 30.39 ± 4.93 | 0.157 |
| Gravidity | | | |
| Primigravida | 32 (50.8) | 31 (49.2) | 0.090 |
| Multigravida | 63 (38.4) | 101 (61.6) | |
| Type of complaint | | | |
| Maternal | 31 (51.7) | 29 (48.3) | 0.072 |
| Neonatal | 64 (38.3) | 103 (61.7) | |
| Type of hospital | | | |
| Private hospital | 37 (43.0) | 49 (57.0) | 0.904 |
| Governmental hospital | 29 (39.7) | 44 (60.3) | |
| Teaching hospital | 29 (42.6) | 39 (57.3) | |
| Type of delivery | | | |
| NVD | 49 (42.2) | 67 (57.7) | 0.080 |
| Elective C/S | 8 (72.7) | 3 (27.2) | |
| Emergency C/S | 30 (42.2) | 41 (57.7) | |
| C/S with indication | 8 (27.6) | 21 (72.4) | |
| Accuser | | | |
| Gynecologist | 43 (31.8) | 72 (62.6) | 0.168 |
| Other medical staffs | 52 (46.4) | 60 (53.6) | |
| Instrumental delivery | | | |
| Positive | 11 (68.8) | 5 (31.2) | **0.021** |
| Negative | 38 (38.0) | 62 (62.0) | |
| The reasons for lawsuit | | | |
| Maternal death | 16 (84.2) | 3 (15.8) | **< 0.001** |
| Neonatal death or IUFD | 33 (41.8%) | 46 (58.2) | |
| Neonatal asphyxia | 23 (54.8%) | 19 (45.2) | |
| Maternal organ damage | 6 (28.6) | 15 (71.4) | |
| Birth trauma | 8 (19.5) | 33 (80.5) | |
| Post procedure complication | 7 (43.8) | 9 (56.2) | |
| Neonatal RDS | 0 | 3 (100) | |
| Other | 2 (33.3) | 4 (66.7) | |

C/S: caesarean section, NVD: normal vaginal delivery, IUFD: intra uterine fetal death, RDS: Respiratory distress syndrome

even after adjusting for variables such as maternal age, type of hospital, reason for the lawsuit, and time interval between delivery and filing of claims (P = 0.020, B = 6.18 (1.33–28.65)).

The mean age of women who had claims associated with a caesarean section was significantly higher (31.7 ± 4.8) compared to women who had claims after a vaginal delivery (30.2 ± 5.4) (P = 0.038). Furthermore, the rate of claims related to maternal complications was significantly higher among women who underwent a caesarean section (45 cases, 40.5%) compared to those who had a vaginal delivery (15 cases, 12.9%) (P < 0.001).

## Discussion

The study findings indicate a significant number of delivery-related complaints associated with neonatal complications. Specifically, a notable proportion of these complaints were related to perinatal mortality, followed by cases of neonatal asphyxia. Additionally, the study found a higher risk of potential financial implications in maternal claims, although this difference was not statistically significant. Approximately half of the claims were related to obstetrician-gynecologist physicians. Additionally, the majority of claims among other healthcare providers were against midwives, totalling 37 (33%). It is worth noting that factors such as maternal age, type of hospital (private, teaching, or public), and mode of delivery (vaginal delivery or caesarean section) did not have a significant impact on the likelihood of paid claims. However, instrumental delivery was found to significantly increase the risk of malpractice.

Importantly, the study also found that the risk of maternal complications was significantly higher in women who underwent a caesarean section, while the risk of neonatal problems was greater in women who had a vaginal delivery. In the present study, 111 claims (48.9%) were related to caesarean section. However, it should be noted that this does not necessarily imply that caesarean section increases the risk of claims against gynecologists. A recent systematic review indicated that the rate of caesarean section in Iran was approximately 48% [18]. Therefore, the proportion of claims related to caesarean section in this study aligns with the prevalence of caesarean section procedures in the country. Additionally, the rate of paid claims among elective caesarean sections was higher than emergency caesarean section. In light of these findings, it is crucial for gynecologists to be aware of the associated factors and take appropriate measures to mitigate risks, regardless of the delivery method employed.

As previously mentioned, the field of obstetrics and gynecology is known to carry a higher risk of lawsuits in various countries [19, 20]. This holds true in Iran as well, where obstetrics and gynecology-related claims are prevalent and have seen an increase in recent years [21, 22]. It has been observed that obstetrical claims tend to be more common than gynecological complaints [23]. A significant portion of these claims is specifically related to delivery events [24].

In a previous study conducted by Azimi et al. in Iran, they conducted an observational study on 1,116 medical claims in the field of obstetrics and gynecology over a two-year period [21]. The study revealed that approximately half of all claims were directly linked to obstetric complications. Interestingly, their findings are in line with our own, as Azimi et al. also found that around 50% of the claims implicated obstetricians and gynecologists, while the remaining cases involved accusations against other medical staff [21].

According to this study, approximately 60% of all claims were successfully defended. These findings align with a recent study in the Saudi Arabia [2]. Another study conducted in Italy revealed that approximately 30% of complaints related to childbirth resulted in payment compensation [6]. In consistent with our results, the majority of these claims were associated with caesarean section complications, extending beyond the surgical procedure itself. They also highlighted the usefulness of using medico-legal documents to inform specialists about the risks of to mitigate them [6, 25].

The study findings indicate that most claims against gynecologists did not result in payment. However, it was reported that the process of lawsuit prosecution still required significant time and effort from the gynecologists to defend themselves in multiple sessions [26, 27]. This legal process had negative psychological effects on the gynecologists involved [28]. To mitigate the risk of medicolegal issues and complaints, it was recommended that appropriate documentation be maintained during procedures, particularly in emergency situations. Additionally, patients should be fully informed and provide their consent [29, 30].

In accordance with our findings, a case-control study comparing 290 claims related to complications during delivery with 262 control cases showed that there were no significant differences in maternal age, delivery setting, and mode of delivery between the two groups [31].

Another study, which analyzed approximately 11,000 medico-legal complaints over a 10-year period, found that gynecologists faced a substantially higher liability for payment and legal prosecution compared to the majority of other specialists. Furthermore, in line with our findings, the study indicated that the majority of complaints resulting in payments were associated with the utilization of vacuum extraction [32].

Previous studies have consistently demonstrated that the fear of healthcare providers regarding the risk of medicolegal lawsuits has a significant impact on the rate of caesarean sections. This pressure often leads to a reduced likelihood of performing primary vaginal deliveries and vaginal births after caesarean sections (VBAC), ultimately contributing to an overall increase in the caesarean section rate [20, 33, 34]. A study conducted in a Florida hospital supported these findings by revealing that obstetrician-gynecologists with a higher proportion of caesarean deliveries experienced a lower occurrence of malpractice incidents [35]. However, it is important to note that another study by Kim et al. found no association between the prevalence of caesarean sections and the fear of malpractice [36].

The strength of this study is that the information derived from medicolegal documents is invaluable for OB/GYN residents and specialists, as they are frequently named defendants in malpractice claims and face substantial financial liabilities each year.

However, it is important to acknowledge the limitations of our study. Firstly, while we examined the claims, we did not evaluate the total number of each delivery type during this period. Further studies are needed to determine the proportion of claims relative to the overall number of each delivery type in hospitals and to compare the claim rates with the total number of deliveries to validate our findings. Additionally, another limitation is the lack of investigation into the reasons behind claims related to delivery from the perspectives of healthcare providers and other stakeholders.

## Conclusion

In summary, this study revealed that maternal complications were notably more frequent in caesarean section cases, while neonatal claims were more commonly observed in vaginal deliveries. Furthermore, instrumental delivery was conducted in 13.8% of vaginal deliveries and was identified as a risk factor for malpractice claims.. As the majority of claims do not result in paid claims, it is recommended to enhance public information to reduce the risk of lawsuits against OB/GYN physicians.

## Supporting information

**S1 Data.**
(SAV)

## Author Contributions

**Conceptualization:** Zeinab Mansouri, Sedigheh Hantoushzadeh.

**Data curation:** Zahra Shabannejad, Elham Bazmi, Mehdi Foroozesh.

**Formal analysis:** Nasim Eshraghi, Marjan Ghaemi.

**Investigation:** Zahra Shabannejad, Elham Bazmi, Sepideh Azizi.

**Supervision:** Zeinab Mansouri, Sedigheh Hantoushzadeh.

**Validation:** Zahra Shabannejad, Elham Bazmi.

**Writing – original draft:** Nasim Eshraghi, Mehdi Foroozesh, Mohammad Haddadi.

**Writing – review & editing:** Marjan Ghaemi, Mohammad Haddadi, Sepideh Azizi.

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
