## [Decision Letter · Decision Letter 0]

26 Apr 2024

PONE-D-24-07510Analysis of Medico-Legal Claims Related to Deliveries: Cesarean Section vs. Vaginal DeliveryPLOS ONE

Dear Dr. Hantoushzadeh,

Thank you for submitting your manuscript to PLOS ONE. After careful consideration, we feel that it has merit but does not fully meet PLOS ONE’s publication criteria as it currently stands. Therefore, we invite you to submit a revised version of the manuscript that addresses the points raised during the review process. The article needs structural, formal, and clarity improvements (as indicated by the reviewers). Moreover, is necessary an editing withe typos and grammar check (if possible, use professional editing services).

We look forward to receiving your revised manuscript.

Kind regards,

Andrea Cioffi

Academic Editor

PLOS ONE

Journal Requirements:

2. Thank you for submitting the above manuscript to PLOS ONE. During our internal evaluation of the manuscript, we found significant text overlap between your submission and previous work in the [introduction, conclusion, etc.].

Please revise the manuscript to rephrase the duplicated text, cite your sources, and provide details as to how the current manuscript advances on previous work. Please note that further consideration is dependent on the submission of a manuscript that addresses these concerns about the overlap in text with published work.

[If the overlap is with the authors’ own works: Moreover, upon submission, authors must confirm that the manuscript, or any related manuscript, is not currently under consideration or accepted elsewhere. If related work has been submitted to PLOS ONE or elsewhere, authors must include a copy with the submitted article. Reviewers will be asked to comment on the overlap between related submissions (http://journals.plos.org/plosone/s/submission-guidelines#loc-related-manuscripts).]

We will carefully review your manuscript upon resubmission and further consideration of the manuscript is dependent on the text overlap being addressed in full. Please ensure that your revision is thorough as failure to address the concerns to our satisfaction may result in your submission not being considered further.

3. In this instance it seems there may be acceptable restrictions in place that prevent the public sharing of your minimal data. However, in line with our goal of ensuring long-term data availability to all interested researchers, PLOS’ Data Policy states that authors cannot be the sole named individuals responsible for ensuring data access (http://journals.plos.org/plosone/s/data-availability#loc-acceptable-data-sharing-methods).

Reviewers' comments:

Reviewer's Responses to Questions

**Comments to the Author**

1. Is the manuscript technically sound, and do the data support the conclusions?

Reviewer #1: Partly

Reviewer #2: Yes

2. Has the statistical analysis been performed appropriately and rigorously? 

Reviewer #1: No

Reviewer #2: Yes

3. Have the authors made all data underlying the findings in their manuscript fully available?

Reviewer #1: Yes

Reviewer #2: Yes

4. Is the manuscript presented in an intelligible fashion and written in standard English?

Reviewer #1: Yes

Reviewer #2: No

5. Review Comments to the Author

Reviewer #1: Though the authors cover a pertinent topic on medico-legal claims, the write-up needs to be improved to ensure that the grammar is right and appropriate. The manuscript needs to make a number of corrections to make it more informative to the readership. The statistical analysis needs to be improved. A table of baseline characteristics needs to be added and preferably adding a multivariate analysis would improve the strength of the associations made by the authors. The discussion is more around Europe than even the local context or SSA. The conclusions don't seem to fully arise from the study and the recommendations drawn are not from the study findings.

Reviewer #2: Dear authors, thank you for conducting an interesting topic; the findings have important implications for clinicians and public health. However, I have some comments that should be addressed before the publication

- the entire manuscripts need to be revised for grammatical errors and punctuation issues and references for some sentences are missing ( e.g. Discussion part paragraph 2 the end sentences should be added.

- The authors should include statistical analysis methods in the abstract part

- the authors in the main manuscript, stated the conducted using medico-legal dispute documents from March 2018 to February 2020. This information is better if also available in the methods section of the abstract

- also, there are some inconsistencies in your abstract and main manuscript. for example, in the abstract, we analyzed medico-legal cases that during two years, but in your result we stated .. a total of 227 women with obstetric claims, meeting the inclusion criteria, were included in this study over three years. why does difference happen? study period should be consistent

- this study does not incorporate a qualitative study of the reason for claims associated with delivery from a Health care provider perspective and mothers, and other stakeholders. so, better if should be stated as the limitation of this study.

- finally. based on your findings why not suggest that public intervention should be targeted at reducing the prevalence of delivery-related claims?

6. PLOS authors have the option to publish the peer review history of their article (what does this mean?). If published, this will include your full peer review and any attached files.

Reviewer #1: No

Reviewer #2: **Yes: **Ebisa Zerihun

---

## [Author Response · Author response to Decision Letter 0]

7 May 2024

Dear Editor and Reviewers,

I would like to express my gratitude for considering this manuscript for revision and for providing valuable comments. I have carefully addressed each comment and made the necessary changes in the main text accordingly.

Thank you once again for your time and input. I look forward to your further evaluation of the revised manuscript.

Sincerely

Journal Requirements:

Thank you. I ensured that manuscript meets PLOS ONE's style requirements

2. Thank you for submitting the above manuscript to PLOS ONE. During our internal evaluation of the manuscript, we found significant text overlap between your submission and previous work in the [introduction, conclusion, etc.].

Please revise the manuscript to rephrase the duplicated text, cite your sources, and provide details as to how the current manuscript advances on previous work. Please note that further consideration is dependent on the submission of a manuscript that addresses these concerns about the overlap in text with published work.

[If the overlap is with the authors’ own works: Moreover, upon submission, authors must confirm that the manuscript, or any related manuscript, is not currently under consideration or accepted elsewhere. If related work has been submitted to PLOS ONE or elsewhere, authors must include a copy with the submitted article. Reviewers will be asked to comment on the overlap between related submissions (http://journals.plos.org/plosone/s/submission-guidelines#loc-related-manuscripts).]

We will carefully review your manuscript upon resubmission and further consideration of the manuscript is dependent on the text overlap being addressed in full. Please ensure that your revision is thorough as failure to address the concerns to our satisfaction may result in your submission not being considered further.

Thank you for your feedback. I have thoroughly reviewed the entire manuscript using the plagiarism detection tool, iThenticate. I have also attached the file for your reference. Based on the results, I believe that the similarity rate is within an acceptable range, considering that there may be some common words or phrases present in the manuscript. If you have any specific concerns or questions about the detected similarities, please let me know, and I'll be happy to assist you further.

3. In this instance it seems there may be acceptable restrictions in place that prevent the public sharing of your minimal data. However, in line with our goal of ensuring long-term data availability to all interested researchers, PLOS’ Data Policy states that authors cannot be the sole named individuals responsible for ensuring data access (http://journals.plos.org/plosone/s/data-availability#loc-acceptable-data-sharing-methods).

Thank you. I provide the information for a non-author contact person who can be reached for data access. 

Name: Zahra Omidi

Email: z.omidii67@gmail.com

Tel: +98 9123491650

Thank you. I provided the orchid id for the corresponding authors.

Is the manuscript technically sound, and do the data support the conclusions?

Reviewer #1: Partly

Reviewer #2: Yes

2. Has the statistical analysis been performed appropriately and rigorously?

Reviewer #1: No

Reviewer #2: Yes

3. Have the authors made all data underlying the findings in their manuscript fully available?

Reviewer #1: Yes

Reviewer #2: Yes

4. Is the manuscript presented in an intelligible fashion and written in standard English?

Reviewer #1: Yes

Reviewer #2: No

5. Review Comments to the Author

Reviewer #1: Though the authors cover a pertinent topic on medico-legal claims, the write-up needs to be improved to ensure that the grammar is right and appropriate. The manuscript needs to make a number of corrections to make it more informative to the readership. 

Thank you for your comment. I have revised the manuscript to address the grammar issues.

The statistical analysis needs to be improved. A table of baseline characteristics needs to be added and preferably adding a multivariate analysis would improve the strength of the associations made by the authors. 

Thank you for your comment. I have made the following revisions to the manuscript:

1. Added the baseline characteristics in Table 1.

2. Conducted a multivariable analysis to enhance the strength of the study.

3. Included the results of the regression analysis in the Results section.

The discussion is more around Europe than even the local context or SSA. 

Thank you for your comment. I have revised the discussion part of the manuscript to include the results of relevant Iranian research.

The conclusions don't seem to fully arise from the study and the recommendations drawn are not from the study findings.

Thank you for your comment. I have revised the conclusion part. Regarding the recommendation, given the high number of lawsuits against OB/GYN surgeons and the importance of identifying the factors associated with these claims, our study's findings suggest that evaluating medicolegal documents could assist surgeons in being better informed and potentially reducing the risk of receiving complaints.

Reviewer #2: Dear authors, thank you for conducting an interesting topic; the findings have important implications for clinicians and public health. However, I have some comments that should be addressed before the publication

- the entire manuscripts need to be revised for grammatical errors and punctuation issues and references for some sentences are missing ( e.g. Discussion part paragraph 2 the end sentences should be added.

Thank you for your comment. I appreciate your review of this article and your interest in the topic. I have added the end sentence to the end of the second paragraph and made revisions to the manuscript accordingly.

- The authors should include statistical analysis methods in the abstract part

Thank you for your comment. I have included the statistical analysis in the abstract part of the manuscript.

- the authors in the main manuscript, stated the conducted using medico-legal dispute documents from March 2018 to February 2020. This information is better if also available in the methods section of the abstract

Thank you for your comment. I have included this information in the abstract part of the manuscript.

- also, there are some inconsistencies in your abstract and main manuscript. for example, in the abstract, we analyzed medico-legal cases that during two years, but in your result we stated .. a total of 227 women with obstetric claims, meeting the inclusion criteria, were included in this study over three years. why does difference happen? study period should be consistent

Thank you for your comment. I apologize for the mistake. I have corrected it.

- this study does not incorporate a qualitative study of the reason for claims associated with delivery from a Health care provider perspective and mothers, and other stakeholders. so, better if should be stated as the limitation of this study.

Thank you for your comment. I have incorporated this limitation into our study.

- finally. based on your findings why not suggest that public intervention should be targeted at reducing the prevalence of delivery-related claims?

Thank you for your comment. I have incorporated this conclusion into our study.

---

## [Decision Letter · Decision Letter 1]

5 Jun 2024

PONE-D-24-07510R1Analysis of Medico-Legal Claims Related to Deliveries: Cesarean Section vs. Vaginal DeliveryPLOS ONE

Dear Dr. Hantoushzadeh,

Thank you for submitting your manuscript to PLOS ONE. After careful consideration, we feel that it has merit but does not fully meet PLOS ONE’s publication criteria as it currently stands. Therefore, we invite you to submit a revised version of the manuscript that addresses the points raised during the review process. Formal and structurally relevant revisions are still needed (as indicated by one of the reveiwers). Please pay particular attention to the reviewer’s comments.

We look forward to receiving your revised manuscript.

Kind regards,

Andrea Cioffi

Academic Editor

PLOS ONE

Reviewers' comments:

Reviewer's Responses to Questions

**Comments to the Author**

1. If the authors have adequately addressed your comments raised in a previous round of review and you feel that this manuscript is now acceptable for publication, you may indicate that here to bypass the “Comments to the Author” section, enter your conflict of interest statement in the “Confidential to Editor” section, and submit your "Accept" recommendation.

Reviewer #1: All comments have been addressed

Reviewer #2: (No Response)

2. Is the manuscript technically sound, and do the data support the conclusions?

Reviewer #1: No

Reviewer #2: (No Response)

3. Has the statistical analysis been performed appropriately and rigorously? 

Reviewer #1: No

Reviewer #2: (No Response)

4. Have the authors made all data underlying the findings in their manuscript fully available?

Reviewer #1: Yes

Reviewer #2: (No Response)

5. Is the manuscript presented in an intelligible fashion and written in standard English?

Reviewer #1: No

Reviewer #2: (No Response)

6. Review Comments to the Author

Reviewer #1: Reviewer’s comments: Dr Herbert Kayiga

General comments:

I take this privilege to appreciate authors for revising the manuscript. Most of the comments were not addressed in the current submission. I wonder whether they submitted the correct version of manuscript. Otherwise, I don’t see any improvement made in the revised version.

Specific comments:

Abstract:

Introduction: It would be ideal to mention the authority like the WHO, FIGO or Iranian Ministry of Health said …. rather than “it was said…. I still notice the authors didn’t revise this aspect. The write up on “Chi-square or t-tests were employed to compare factors between paid claims and closed claims” should be put in the methods section.

Methods: The revision has been made concerning the study period but other aspects as indicated in the reviewer’s report are not made.

See the comments before

It would be prudent to indicate where exactly the study was conducted and the setting “was it in private or public health facility setting?” Who was responsible for making the claims? Are there charges implicated with the medico-legal claims? What kind of penalties are available for the implicated healthcare providers? Where there any instrumental deliveries as indicated in the settings? If yes, what proportion of women had instrumental delivery?

Results:

It’s still quite confusing as the comments raised in the first report are not addressed. Kindly see the points below. It’s not clear whether the reference cadre were surgeons or general doctors. What claims were reported against other staffs? Clarity is needed for closed vs paid claims.

Previous comments;

I think it would have been so handy knowing how high the claims that accrued when compared to the vaginal delivery. This is however, implied. What was the reference group? What kind of neonatal complications led to the observed high claims as seen in the vaginal route? What other cadres made up for the other staffs? Are these cadres permitted to perform the caesarean sections as the obstetricians? Elaborate more on the difference between closed and paid claims to avoid confusion.

The authors mention “However, the number of instrumental deliveries and the reasons for claims, exhibited significant differences between the two groups (P=0.021)”. It’s not clear though the groups that are being compared. How many instrumental deliveries occurred in the caesarean vs. vaginal groups if I can assume that these are the comparative groups?

The conclusion: I notice that comments were still not addressed.

“The study concluded that caesarean section did not result in an increased risk of medicolegal claims or payment compensation compared to vaginal delivery”, doesn’t appear to be arising from the work. If caesarean section constituted high maternal complications than vaginal delivery, why is it then that it didn’t have higher claims? What complication rates were sufficient to constitute a claim?

Introduction/Background:

I notice the comments raised are not addressed as expected. I wonder whether authors really saw the comments in this area. The global C/s rate is not mentioned. The role of other staffs is not clear and how they get implicated is not clear. More clarity is needed on the contextual setting in the private and public facilities in Iran in regards to the delivery processes.

See comments below;

Is it the global perspective that all obstetricians/gynaecologists face higher medico-legal claims more other physicians or a context-based observation? In some countries, surgeons are more at risk than the obstetricians.

I think to would be prudent to mention the context or setting of where the reduced mortality following c/sections is observed because in low resource settings, the cost and mortality is way higher following caesarean delivery than observed for vaginal delivery.

What is the global caesarean section rate? I would be nice if wherever there’s a comparison, the reference range is mentioned.

What is the setting like in Iran? Are there any other cadres allowed to do caesarean sections other than obstetricians? Who conducts vaginal deliveries? Is it still the obstetricians or even the midwives do the job? I would be ideal helping the readership to have glimpse of the delivery processes in Iran. Are the claims similar for elective vs. emergent caesareans? It would have been nice to know of the implications of these medico-legal claims in Iran. Try to introduce the concept of paid and closed claims as indicated in the abstract highlighting the difference.

What authority determines the nature of claim that arises? Is it the legal fraternity or the medical council? It is so informative if you can help the readership view the impact through the lenses of the Iranian medico-legal environment.

Methods: Comments not addressed. I pray the authors submitted the right version for the review! No effort was made to address the concerns raised.

Comments submitted before:

I think it’s best to call it methods rather than method. Please add these years in the abstract write up. It would be nice to separate the study design and the ethical approvals. Write these subsections differently and add more details in each of them.

Who were the participants? Was it the patients or the affected family members? Was it a document review process or you conducted participant interviews? How did you minimize recall biases since the interviews or document review was 3-5 years after the incidences? How did you identify the potential participants? What measures were put in place to ensure that patient safety was observed? At what point were the discussions held? Was it after the hospital admissions or after discharge from the health facilities? What is the setting of the deliveries? Who takes the mandate to determine whether instrumental, vaginal or caesarean section is to be taken? How do the patients get to obtain care in the Iranian setting? Is at the jurisdiction of the Ministry of Health or the patients? Who determines whether the care accorded to the patient is optimal or not so as to determine to proceed to the next phase of making a legal claim?

What was the baseline from which these documents were drawn? I see the other cadres constituted anaesthetists, and midwives. These all have different responsibilities in hospital setting. Did you expect them to have the same weight in terms of claims? It also appears like the cases were obstetric in nature, what kind of gynaecological causes attracted attention in the study period? Were there any other physicians like GPs, Surgeons who had cases in this study period because some of the patients had bowel or bladder injuries?

Kindly add a write up on the study settings. It’s not clear who made the decisions on the wards, who runs the care of the patients. Are the claims more in private than the public settings? Is there any correlation between the patient volumes and the legal claims? Who determine the nature of the claims? How does a medical case become a medico-legal claim? Who files the claims? Is it the patients, their lawyers or the hospital that files a claim? How do the cases get up in court? What kind of penalties are issues in court?

The inclusion and exclusion criteria are not mentioned in the write up. Clarity is needed to determine the proportion of deliveries that turned out to be claims in the study period.

Results: Authors didn’t revise the manuscript as expected. There are typos that need to be revised in the submitted version. There are overlapping tables on subsequent pages like 2. The titles could be improved to have ‘claims’ rather than groups! Significant findings with p-values < 0.05 can be made bold.

Comments raised in previous review

I think it would nice to write more about the settings of the health facilities in the methods rather than introducing them in the result section for the first time. It’s still not clear if it’s a document review or participant interviews. Please add a table to show the baseline characteristics of the participants. It’s not so clear who the other medical staffs that were involved in the study. What participant characteristic were more likely to make a claim? Prime gravida or multiparous? Best in the introduction of the results section to mention how many deliveries occurred generally, how many SVDs, instrumental deliveries (vacuum or forceps) occurred in the study periods? Who assisted in the deliveries and the key aspects that contributed to a claim and a non-claim? How many closed or paid claims were reported? Were there any c/sections that were done with no indication? I don’t see any IUFD in the table? Did you regard perinatal mortality as IUFDs? Otherwise, how did you capture early neonatal deaths? The two groups being compared that were mentioned to be similar are quite confusing. Did you mean among those that malpractice vs. cases with no malpractice or vaginal vs. c/section?

What characteristics were reported among those whose claims were paid? Are these characteristics similar among those who had vaginal vs. caesarean delivery? Adding a multivariate analysis would have improved the overall impact and eliminate any confounding factors.

Discussion: Authors haven’t revised the manuscript as expected.

Comments from previous review:

It would be ideal to recap the overall aim of the study and mention the key significant findings before comparison with the other studies. Knowing the contribution of the other specialties in regards to the claims contributed by obs/gyn would have been so handy at this point. In the write up there are no gyn cases! So how would you conclude that obs has more legal claims than gyn even in the Iranian setting? Which region are you referring to with the CS rate of 48%? Please don’t assume that the readers see through your eyes.

Mentioning “Additionally, the rate of paid claims among emergency caesarean sections was similar to that of vaginal deliveries” is not the best comparison. I think I would compare it to the elective cases as the participant characteristics would be different”. I find most of the comparisons in Europe! It would have been interesting to hear how neighbouring countries to Iran or Sub-Saharan Africa would compare in regards to medico-legal claims and obstetric cases. The strength of the study is not water-tight. Having it conducted in Iran doesn’t make it a strength in my view. Is it that there are many claims following c/s in Iran or because WHO thinks the C/s rate is higher than in other settings?

The limitations are not so clear as well. How would conducting a case-control study improve the study findings for your study? What did you mean by conduction the whole province evaluation would be a limitation? What challenges did you meet in executing this study?

Conclusion: Authors haven’t revised the manuscript as per the expectation.

Comments from previous review

In my evaluation, the claim that “this study revealed that claims related to deliveries were often associated with perinatal death” can not be drawn from the study findings. Though instrumental delivery is claimed to be a cause of malpractice, the authors make no mention of how many instrumental deliveries were reported among the claims. Clarity needs to be made in this regard. Though the authors claim to have conducted the study to educate the specialists on medico-legal claims, there is no mention that there was a gap in the knowledge among the specialists. I request for recommendations that can be drawn from the study findings that can inform policy and future research.

Reviewer #2: I am writing to thank you for your careful consideration of my comments. I am pleased to see that you have addressed all of my concerns. Thank you again for your hard work and dedication.

7. PLOS authors have the option to publish the peer review history of their article (what does this mean?). If published, this will include your full peer review and any attached files.

Reviewer #1: **Yes: **Dr. Herbert Kayiga

Reviewer #2: **Yes: **Ebisa Zerihun

---

## [Author Response · Author response to Decision Letter 1]

17 Jun 2024

Dear Reviewer:

Thank you so much for reviewing this article and for your comments. I want to sincerely apologize that I missed addressing your feedback in the previous version. To be honest, I was unable to see your comments in the prior decision letter, which led to this oversight on my part. I feel terrible about that, and I have made every effort to thoroughly incorporate your feedback in this latest revision.

Please let me know if there are any other changes or clarifications you would like me to make. I am looking forward to hearing from you. I greatly appreciate the time and consideration you have provided throughout this process.

---

## [Decision Letter · Decision Letter 2]

13 Aug 2024

PONE-D-24-07510R2Analysis of Medico-Legal Claims Related to Deliveries: Cesarean Section vs. Vaginal DeliveryPLOS ONE

Dear Dr. Hantoushzadeh,

Thank you for submitting your manuscript to PLOS ONE. After careful consideration, we feel that it has merit but does not fully meet PLOS ONE’s publication criteria as it currently stands. Therefore, we invite you to submit a revised version of the manuscript that addresses the points raised during the review process. Some minor revisions are still needed as indicated by reviewers.

We look forward to receiving your revised manuscript.

Kind regards,

Andrea Cioffi

Academic Editor

PLOS ONE

Journal Requirements:

Reviewers' comments:

Reviewer's Responses to Questions

**Comments to the Author**

1. If the authors have adequately addressed your comments raised in a previous round of review and you feel that this manuscript is now acceptable for publication, you may indicate that here to bypass the “Comments to the Author” section, enter your conflict of interest statement in the “Confidential to Editor” section, and submit your "Accept" recommendation.

Reviewer #1: All comments have been addressed

Reviewer #3: (No Response)

2. Is the manuscript technically sound, and do the data support the conclusions?

Reviewer #1: Yes

Reviewer #3: No

3. Has the statistical analysis been performed appropriately and rigorously? 

Reviewer #1: Yes

Reviewer #3: Yes

4. Have the authors made all data underlying the findings in their manuscript fully available?

Reviewer #1: Yes

Reviewer #3: Yes

5. Is the manuscript presented in an intelligible fashion and written in standard English?

Reviewer #1: Yes

Reviewer #3: Yes

6. Review Comments to the Author

Reviewer #1: Reviewer’s comments: Dr Herbert Kayiga

General comments:

I take this privilege to appreciate authors for revising the manuscript. Most of the comments have been addressed in the current submission. Minor changes are required to ensure consistence in the write up. The grammar needs to be improved and very long sentences shortened. Add more references to the discussion. It’s more of what you found and less of how your findings compare to existing literature.

Specific comments:

Abstract:

Introduction: We can say that “The Iranian National Health Service (NHS) suggested…. Be consistent with the writing “caesarean section”. In some places you write ‘Cesarean

sections,”

Introduction/Background:

Kindly indicate year the global caesarean rate was taken.

Discussion: The write up has greatly improved but more references need to be added to beef up arguments. For example

“In a previous study conducted by Azimi et al. in Iran, they conducted an observational study on 1,116 medical claims in the field of obstetrics and gynecology over a two-year period”.

“Another study conducted in Italy revealed that approximately 30% of complaints related to childbirth resulted in payment

compensation. In consistent with our results, the majority of these claims were associated with caesarean section complications, extending beyond the surgical procedure itself”.

No references added.

Reviewer #3: Researchers have conducted a suitable study for legal issues and complaints about cesarean section and vaginal delivery.

Such a conclusion cannot be drawn from the results of this study, and the findings do not match the conclusions.

The limitations of the study are not mentioned in detail.

Researchers should note that significant statistical difference and significant statistical significance always mean clinically significant difference and may not be clinically significant. It seems that this happened in this study.

7. PLOS authors have the option to publish the peer review history of their article (what does this mean?). If published, this will include your full peer review and any attached files.

Reviewer #1: **Yes: **Dr Herbert Kayiga

Reviewer #3: No

---

## [Author Response · Author response to Decision Letter 2]

6 Sep 2024

Reviewer #1: Reviewer’s comments: Dr Herbert Kayiga

General comments:

I take this privilege to appreciate authors for revising the manuscript. Most of the comments have been addressed in the current submission. Minor changes are required to ensure consistence in the write up. The grammar needs to be improved and very long sentences shortened. Add more references to the discussion. It’s more of what you found and less of how your findings compare to existing literature.

Thank you so much for your Comments. I sincerely appreciate your valuable comments, which I believe have greatly helped to improve our study. I want to express my deep gratitude for your time and consideration.

I have revised the text and included additional relevant references.

Specific comments:

Abstract:

Introduction: We can say that “The Iranian National Health Service (NHS) suggested….

Thank you for your comment. I revised it. 

 Be consistent with the writing “caesarean section”. In some places you write ‘Cesarean

sections,”

Thank you for your comment. I revised it. 

Introduction/Background:

Kindly indicate year the global caesarean rate was taken.

Thank you for your comment. I added the year.

Discussion: The write up has greatly improved but more references need to be added to beef up arguments. For example

“In a previous study conducted by Azimi et al. in Iran, they conducted an observational study on 1,116 medical claims in the field of obstetrics and gynecology over a two-year period”.

“Another study conducted in Italy revealed that approximately 30% of complaints related to childbirth resulted in payment

compensation. In consistent with our results, the majority of these claims were associated with caesarean section complications, extending beyond the surgical procedure itself”.

No references added.

Thank you for your comment. I added the references.

Reviewer #3: Researchers have conducted a suitable study for legal issues and complaints about cesarean section and vaginal delivery.

Such a conclusion cannot be drawn from the results of this study, and the findings do not match the conclusions.

Thank you for your comment. I revised the conclusion.

The limitations of the study are not mentioned in detail.

Thank you for your comment. I mentioned more detail for limitation. 

Researchers should note that significant statistical difference and significant statistical significance always mean clinically significant difference and may not be clinically significant. It seems that this happened in this study.

Thank you for your comment. Our findings indicate that instrumental delivery is associated with an increased risk of paid claims, a trend also observed in previous studies. While this may evolve in future research, it could be clinically relevant.

---

## [Decision Letter · Decision Letter 3]

10 Oct 2024

Analysis of Medico-Legal Claims Related to Deliveries: Cesarean Section vs. Vaginal Delivery

PONE-D-24-07510R3

Dear Dr. Hantoushzadeh,

We’re pleased to inform you that your manuscript has been judged scientifically suitable for publication and will be formally accepted for publication once it meets all outstanding technical requirements.

Kind regards,

Andrea Cioffi

Academic Editor

PLOS ONE

Additional Editor Comments (optional):

Reviewers' comments:

Reviewer's Responses to Questions

**Comments to the Author**

1. If the authors have adequately addressed your comments raised in a previous round of review and you feel that this manuscript is now acceptable for publication, you may indicate that here to bypass the “Comments to the Author” section, enter your conflict of interest statement in the “Confidential to Editor” section, and submit your "Accept" recommendation.

Reviewer #3: All comments have been addressed

2. Is the manuscript technically sound, and do the data support the conclusions?

Reviewer #3: Yes

3. Has the statistical analysis been performed appropriately and rigorously? 

Reviewer #3: Yes

4. Have the authors made all data underlying the findings in their manuscript fully available?

Reviewer #3: Yes

5. Is the manuscript presented in an intelligible fashion and written in standard English?

Reviewer #3: Yes

6. Review Comments to the Author

Reviewer #3: (No Response)

7. PLOS authors have the option to publish the peer review history of their article (what does this mean?). If published, this will include your full peer review and any attached files.

Reviewer #3: No

---

## [Editor Report · Acceptance letter]

8 Nov 2024

PONE-D-24-07510R3 

PLOS ONE

Dear Dr. Hantoushzadeh, 

I'm pleased to inform you that your manuscript has been deemed suitable for publication in PLOS ONE. Congratulations! Your manuscript is now being handed over to our production team.

Kind regards, 

on behalf of

Dr. Andrea Cioffi 

Academic Editor

PLOS ONE